# Characterization and Safety Evaluation of Autoclaved Gut Commensal *Parabacteroides goldsteinii* RV-01

**DOI:** 10.3390/ijms252312660

**Published:** 2024-11-25

**Authors:** Tzu-Lung Lin, Wan-Jiun Chen, Chien-Min Hung, Yea-Lin Wong, Chia-Chen Lu, Hsin-Chih Lai

**Affiliations:** 1REVIVEBIO Co., Taipei 115, Taiwan; ivy.lin@revivebio.com.tw; 2Microbio Co., Ltd., Taipei 115, Taiwan; wanjiun.chen@onenessbio.com.tw (W.-J.C.); jimmy.hung@microbio.com.tw (C.-M.H.); barny@microbio.com.tw (Y.-L.W.)

**Keywords:** *Parabacteroides goldsteinii*, gut microbiota, beneficial microorganism, food ingredient, safety, toxicity

## Abstract

Gut commensals play important roles in maintaining the homeostasis of human health. Previous studies indicated that the abundance of *P. goldsteinii* in animal hosts was increased by the administration of prebiotics such as polysaccharides purified from iconic oriental medicinal fungi. Subsequently, *P. goldsteinii* was found to exert beneficial effects on the amelioration of multiple chronic inflammation-associated diseases. Even so, during the process of the development of *P. goldsteinii* as a next-generation probiotic (NGP), care has to be taken when it is used as a functional food ingredient. In this study, we isolated a novel *P. goldsteinii* strain, RV-01, from the feces of a healthy adult and carried out comprehensive analyses of its genomic and phenotypic characteristics. Bioinformatic analysis of *P. goldsteinii* RV-01 revealed the absence of potential virulence genes, as well as the presence of genes and traits potentially beneficial to human health, such as the production of short-chain fatty acids, anti-inflammatory lipopolysaccharides, and zwitterionic capsular polysaccharides, as well as immune regulatory proteins. To circumvent any potential side effects, the *P. goldsteinii* RV-01 was autoclaved before proceeding to the nonclinical safety assessment. The autoclaved *P. goldsteinii* RV-01 retained its anti-inflammatory effect in human colon epithelial cells. In addition to the three genotoxicity assays, 28-day subacute and 90-day subchronic animal toxicity studies (the highest dose tested was equivalent to 8.109 × 10^10^
*P. goldsteinii* RV-01 cells/kg body weight/day) were also implemented. The results of all studies were negative for toxicity. These results support the conclusion that autoclaved *P. goldsteinii* RV-01 is safe for use as a food ingredient.

## 1. Introduction

The homeostasis of the gut microbiota is essential for the maintenance of animal hosts’ health. Among these microbiota, *Parabacteroides* spp. are core members of the human gut microbiota and have an average abundance of 1.27% in humans from 12 populations [1,2]. Parabacteroides have been reported to have the physiological characteristics of carbohydrate metabolism, and they secrete short-chain fatty acids [1]. In particular, recent studies of *P. goldsteinii* have shown its beneficial effects in animal hosts, demonstrating that *P. goldsteinii* has great potential for development as a next-generation probiotic (NGP) to help with the amelioration of chronic inflammation-related diseases [1,3]. In a choline-deficient, amino acid (CDAA)-defined mouse steatohepatitis model, the abundance of *P. goldsteinii* was decreased [4]. Furthermore, whereas the abundance of *P. goldsteinii* was negatively correlated with obesity-related indexes (fasting glucose, AUC of GTT, body weight gain, serum triglyceride, and LBP), as well as inflammatory factors (TNF-α and IL-1β), it was positively correlated with the serum levels of HDL-C and IL-10 in HFD-fed mice [5].

The identification of *P. goldsteinii* as an NGP was originally inspired by the observation that the administration of purified high-molecular-weight polysaccharides derived from *Ganoderma lucidum* and *Ophiocordyceps sinensis* significantly increased the abundance of *P. goldsteinii* and ameliorated obesity and metabolic syndrome in mice fed with a high-fat diet [6,7]. To address the ameliorative function of *P. goldsteinii*, the clinical isolate strain *P. goldsteinii* (ATCC JCM13446) [8] was used as a single strain to evaluate its potential in reducing obesity and metabolic disorders in mice. Although no adverse effects were identified, oral administration of live *P. goldsteinii* was able to counteract body weight and fat mass gain, as well as glucose intolerance induced by high-fat feeding [7]. Furthermore, daily supplementation with live *P. goldsteinii* was associated with a decrease in the plasmatic level of proinflammatory lipopolysaccharides (LPSs), an increase in intestinal tight junction proteins, and an overall improvement in gut barrier function, which had deteriorated in a mouse model of diet-induced obesity [7]. Subsequently, the commensal *P. goldsteinii* strain MTS01 isolated from the mice was shown to ameliorate the pathology of chronic obstructive pulmonary disease (COPD) [9]. A reduction in intestinal inflammation and the enhancement of cellular mitochondrial and ribosomal activities in the colon, the systematic restoration of aberrant host amino acid metabolism in sera, and the inhibition of lung inflammation act as important ameliorative mechanisms against COPD. In addition, the LPS derived from *P. goldsteinii* is anti-inflammatory and ameliorates COPD by acting as an antagonist of the toll-like receptor 4 (TLR4) signaling pathway [9]. No aberrant effects were observed during this study.

Recently, an increasing number of studies have further revealed that *P. goldsteinii* may function as an NGP associated with or involved in the amelioration of many other chronic inflammation-related diseases [1,2]. These diseases included obesity and metabolic syndrome [10,11], murine colitis [12], *Helicobacter pylori*-induced gastric pathology [13], maternal immune activation-induced autism-relevant behaviors [14], anastomotic healing in colorectal cancer surgery [15], food allergies [16], aspirin-mediated intestinal damage [17], and imiquimod-induced lupus in mice [18], as well as the induction of IL-22 and the subsequent enhanced expression of the antimicrobial peptides Reg3γ and Reg3β via *P. goldsteinii’s* outer membrane protein A1 (OmpA1) [19]. Specifically, *P. goldsteinii* was shown to assist in the maturation and development of the immune system [20,21]. *P. goldsteinii* produced β-hexosaminidase, a conserved enzyme identified in the phylum Bacteroidota, and promoted the development of a subtype of CD4^+^ effector T cells (CD4IELs) that regulate adaptive immunity in the intestinal mucosa, thereby playing an important role in controlling inflammation and maintaining homeostasis of the intestinal mucosa [20]. In a cohort study focusing on the analysis of the relationship between *Parabacteroides* and obesity in different human subpopulations (e.g., with respect to age and sex) and its association with subsequent weight changes, *P. goldsteinii* was negatively correlated with blood pressure, body mass index (BMI), and waist circumference, as well as with the prevalence of obesity [22].

Although no evidence has been reported on the causality of *P. goldsteinii* as a disease pathogen, it is possible to isolate *P. goldsteinii* from clinical samples [8]. Therefore, before *P. goldsteinii* RV-01 can be approved as a food ingredient, a thorough safety assessment should be performed. In this study, we specifically isolated a novel *P. goldsteinii* strain, RV-01, from the feces of a healthy adult. Comprehensive physiological and genotypic analyses, as well as monitoring of immune modulation activity in the intestine of a germ-free mouse mono-colonized with *P. goldsteinii* RV-01, were performed. To circumvent any potentiality that live *P. goldsteinii* may cause unexpected side effects, we autoclaved the sample in order to kill *P. goldsteinii* RV-01 [23]. Subsequently, following the US Food and Drug Administration (FDA) and European Food Safety Authority (EFSA) guidance on toxicity testing required for the safety assessment of a new nonabsorbable food ingredient [24,25], the autoclaved *P. goldsteinii* RV-01 was subjected to nonclinical studies assessing its potential for genotoxicity, its subacute toxicity, and its subchronic toxicity. No adverse effects were identified. Based on these observations, the autoclaved *P. goldsteinii* RV-01 produced by fermentation may be considered as a safe food ingredient.

## 2. Results

### 2.1. Descriptions of P. goldsteinii RV-01

Human gut commensal *P. goldsteinii* is a Gram-negative, rod-shaped, strictly anaerobic, and non-spore-forming bacterium belonging to the Bacteroidota phylum [26]. *P. goldsteinii* is involved in the metabolic regulation of the host [1,7] and grows on medium containing 20% bile [26]. The *P. goldsteinii* commensal strain RV-01 was originally isolated from the feces of a healthy adult. A 16S rDNA full-length sequence analysis was used to identify and group *P. goldsteinii* RV-01, which showed 99.71% identity to the *P. goldsteinii* JCM13446 in the NCBI BLAST analysis. The 16S rRNA DNA sequence of *P. goldsteinii* RV-01 was further analyzed by using the BLAST algorithm against the database of 16S ribosomal RNA sequences (Bacteria and Archaea). The neighbor-joining phylogenetic tree showing the relationship between *P. goldsteinii* RV-01 and similar species and strains indicated that *P. goldsteinii* RV-01 and two other *P. goldsteinii* strains were annotated within one cluster (Figure 1A).

When growing on CDC anaerobic blood agar plates, *P. goldsteinii* RV-01 colonies are smooth and circular, with a diameter of 1–2 mm, gray to off-white–gray, and slightly convex (Figure 1B). The morphology of *P. goldsteinii* RV-01 under a transmission electron microscope is shown in Figure 1C. The RV-01 was inactivated by heat treatment (100 °C for 15 min), autoclaving (121 °C for 15 min), or pasteurization (70 °C for 30 min). The activities of *P. goldsteinii* RV-01 that was inactivated at different temperatures were compared (Appendix A). The cellular TLR2 activation activity of autoclaved RV-01 was higher than that of heat-treated or pasteurized RV-01 and was comparable with that of live RV-01. Therefore, autoclaving was adopted for the production of the *P. goldsteinii* RV-01 ingredient. The *P. goldsteinii* RV-01 ingredient was prepared as an autoclaved, lyophilized, off-white to beige, odorless powder containing more than 1.2 × 10^10^ total fluorescence units (TFUs)/g of powder. Proximate analysis of the ingredient is provided in Table 1. The averaged results of the proximate analyses of five batches showed that *P. goldsteinii* RV-01 powder is mainly composed of carbohydrates (95.1–96.3 g/100 g), protein (2.5–3.0 g/100 g), ash (0.2–0.3 g/100 g), and fat (0.4–0.9 g/100 g).

### 2.2. Genomic and Phenotypic Characteristics for Safety Evaluation

#### 2.2.1. Whole Genomic Characteristics

The whole genome of *P. goldsteinii* RV-01 was sequenced by nanopore genome sequence analysis. The genome sequence and annotation of the *P. goldsteinii* RV-01 strain were deposited at NCBI under BioProject number PRJNA1153404. The completed circular chromosomal map of *P. goldsteinii* RV-01, with a total length of 6,631,096 bps, is shown in Figure 1D. The GC content of the genome is 43%, and no plasmid DNA was detected (Appendix A). The nucleotide-level genomic similarity between the *P. goldsteinii* RV-01 genome and those of two *P. goldsteinii* strains (MTS01 and DSM19448), as well as of four *Parabacteroides* species (*P. distasonis* ATCC8503, *P. merdae* ATCC43184, *P. gordonii* DSM23371, and *P. faecis* DSM102983) was analyzed by using average nucleotide identity (ANI) analysis via the ANI calculator (https://www.ezbiocloud.net/tools/ani, accessed on 28 August 2024) using the OrthoANIu algorithm [27]. In Table 2, we can see that the ANI values between *P. goldsteinii* RV-01 and the two *P. goldsteinii* strains (MTS01 and DSM19448) are 98.17% and 98.44%, respectively. The ANI values between *P. goldsteinii* RV-01 and the four *Parabacteroides* species are less than 85% (ranging between 73.83 and 83.96%).

Annotation of the sequences of *P. goldsteinii* RV-01 revealed that the genome consists of 5241 coding DNA sequences, 79 tRNA genes, and 18 rRNA genes. The functional annotation results of the Clusters of Orthologous Genes (COG) database divided into 22 categories are shown in Figure 1E. Among these, the most abundant category is the signal transduction mechanisms, followed by general function and cell wall/membrane/envelope biogenesis.

#### 2.2.2. Minimum Inhibitory Concentration (MIC) Evaluation of Antibiotics

The MIC results for the antibiotics tested are reported in Table 3. EFSA did not provide the microbiological cut-off values for Parabacteroides species [28]. Therefore, three *P. goldsteinii* strains and two *Parabacteroides* species were included for comparison. The results indicated that the observed antibiotic sensitivity of *B. fragilis* ATCC25285 was in compliance with the MIC QC range. Furthermore, *P. goldsteinii* RV-01 presented high resistance levels to aminoglycosides (gentamycin, kanamycin, streptomycin, and apramycin), vancomycin, nalidixic acid, sulfonamide, and trimethoprim. The MICs obtained for these antimicrobials tested were similar among the three *P. goldsteinii* strains and two *Parabacteroides* species, indicating that resistance to these antimicrobials may be considered intrinsic. *P. goldsteinii* RV-01 was susceptible to the other antibiotics (clindamycin, tetracycline, and chloramphenicol) according to the MIC breakpoints for susceptible (S) (CLSI, 2020). Taken together, the MICs of *P. goldsteinii* RV-01 showed a similar level among the different *P. goldsteinii* strains tested.

#### 2.2.3. Bioinformatic Analyses for Antibiotic Resistance and Potential Virulence genes

Analysis of the *P. goldsteinii* RV-01 genome was conducted to screen for potential antibiotic resistance and virulence factor genes using homology search tools and comparing the results with the Comprehensive Antibiotic Resistance Database (CARD) and Virulence Factor DataBase (VFDB) [29,30]. To identify antimicrobial resistance sequence matches, thresholds of >60% coverage and >70% identity were applied based on relevant EFSA guidance [31]. Only one gene that showed a significant match with the tetracycline resistance gene *tetQ* met the threshold qualifications of >60% coverage and >70% identity (Appendix A). However, intriguingly, the results of the minimal inhibitory concentration (MIC) assay showed that *P. goldsteinii* RV-01 was susceptible to tetracycline (Table 3). There was a possibility that the DNA fragment of the complete *tetQ* gene may still be present in the *P. goldsteinii* RV-01 ingredient. To determine whether the *tetQ* might be destroyed in this ingredient, PCR followed by agarose gel electrophoresis was performed to detect the presence of *tet*Q. The result showed that the *tetQ* gene was not detected in the *P. goldsteinii* RV-01 ingredient (Appendix A), excluding the potential for horizontal spreading of the *tetQ* gene.

In silico analysis of the *P. goldsteinii* RV-01 genome was also conducted to screen for potential virulence factor genes; this analysis used homology search tools and compared the results with the Virulence Factor DataBase (VFDB). Thresholds of >60% coverage and >80% identity for virulence factor identity were applied based on relevant EFSA guidance [31]. No genes in the *P. goldsteinii* RV-01 genome demonstrating identity to known virulence factors were revealed (Appendix A). Together, the bioinformatic analyses indicated that *P. goldsteinii* RV-01 is a non-pathogenic bacterium.

#### 2.2.4. Bioinformatic Analyses for Allergic Proteins

A genomic and bioinformatic analysis was conducted to evaluate the allergic potential of the proteins present in *P. goldsteinii* RV-01. The GenBank *P. goldsteinii* RV-01 genome assembly, containing 5241 protein coding sequences (CDSs), was analyzed for potential allergenic proteins; sequence homology tools in Allermatch (http://allermatch.org, accessed on 30 July 2024) [32] were used for this, via the following three types of analysis: small exact word match, 80-amino-acid sliding window alignment, and full-length alignment (Appendix A).

In the full-length sequence homology search, 129 hits (E-value ≤ 0.001) were identified, of which 33 hits had greater than 35% identity (ranging between 35.1 and 70.6%) (Appendix A). In the 80-amino-acid sliding window alignment, 174 hits with greater than 35% identity were identified (ranging between 35.0 and 86.2%) (Appendix A). An exact match search for six amino acids was also performed. Thirteen matches had an exact match, and more than six amino acids were identified (Appendix A). Among these 13 matches, 11 proteins were also identified in the full-length sequence homology search and 80-amino-acid sliding window alignment. Upon closer evaluation, three proteins (endolase, adenosylhomocysteinase, and glyceraldehyde-3-phosphate dehydrogenase A) had >50% identity in both the full-length sequence homology search and 80-amino-acid sliding window alignment. The important consideration in immunogenicity and cross-reactivity is the protein structure. Therefore, although the sequence homology to known allergens was indicated, *P. goldsteinii* RV-01 was not expected to contain allergenic potential, as the *P. goldsteinii* RV-01 ingredient was prepared as an autoclaved and lyophilized powder, which resulted in protein denaturation. Taken together, *P. goldsteinii* RV-01 should have a low risk of allergenicity.

### 2.3. Bioinformatic Analysis of Genes and Traits Potentially Beneficial to Human Health

#### 2.3.1. Production of Short-Chain Fatty Acids (SCFAs)

The SCFAs acetate, propionate, and butyrate are the main metabolites produced in the colon by bacterial fermentation of dietary fibers and resistant starch and have been shown to provide various health benefits [33]. The production of SCFAs in *P. goldsteinii* RV-01 was determined by mass (MS) analysis of bacterial culture supernatant. The major SCFAs found were acetic and propionic acids; minor amounts of isobutyric acid, 3-Methylbutyric acid, 2-Methylbutyric acid, and formic acid are also produced (Figure 2A). The fermentation of complex carbohydrates leads to the production of SCFAs. Therefore, the carbohydrate utilization enzymes of *P. goldsteinii* RV-01 were analyzed with the DIAMOND search against the Carbohydrate-Active Enzyme (CAZy) database [34,35]. A total of 413 genes (7.9% of genome) encoding CAZy, including glycoside hydrolase (GH), glycosyl transferase (GT), carbohydrate binding module (CBM), carbohydrate esterase (CE), and polysaccharide lyase (PL), were identified (Figure 2B). The major CAZy (a total of 239) was GH, which is a class of enzymes that catalyze the hydrolysis of glycosidic bonds in complex sugars. These results indicated that *P. goldsteinii* RV-01 is a complex carbohydrate-degrading and SCFA-producing bacterium.

#### 2.3.2. *P. goldsteinii* RV-01 Reduced TLR4-Related Inflammation

Unlike the LPS of most Gram-negative bacteria (for example, *E. coli*), which elicit proinflammatory responses, *P. goldsteinii* has been reported to produce an anti-inflammatory LPS (Pg-LPS) that antagonizes TLR4 activity. *E. coli* normally produce a lipid A molecule with six acyl chains [36] (Figure 3A). A BLAST analysis of the whole genome sequence of *P. goldsteinii* RV-01 highlighted that the genes involved in lipid A synthesis lacked LpxM, one ortholog of the acyltransferase genes that is responsible for the addition of the sixth acyl chain of LPS, and it was thus expected to produce hypo-acylated lipid A (Figure 3B). As shown in Figure 3C, autoclaved *E. coli* induced profound inflammation indicated by NF-κB activation in human colon epithelial HCT116 cells. In comparison, autoclaved *P. goldsteinii* RV-01 did not show significant activation activity. Moreover, pretreatment with autoclaved *P. goldsteinii* RV-01 significantly decreased the inflammation induced by autoclaved *E. coli* (Figure 3C). These results suggest that autoclaved *P. goldsteinii* RV-01 reduced the TLR4-related inflammation, where its LPS might be responsible for the TLR4 antagonization activity.

#### 2.3.3. Bioinformatic Analyses for Zwitterionic Capsular Polysaccharide (ZPS)

Polysaccharide A (PSA), a very well-studied zwitterionic polysaccharide (ZPS) from the commensal gut bacteria *Bacteroides fragilis*, may induce anti-inflammatory regulatory T cells and suppress proinflammatory T helper 17 cells [37]. PSA has been shown to protect animals from colitis, encephalomyelitis, colorectal cancer, pulmonary inflammation, and asthma [37]. Previous studies demonstrated that *wcfR* gene homologs encoding for an amino sugar synthetase, as well as *wcfS* gene homologs encoding for an amino sugar transferase, are both needed in the synthesis of ZPS [38,39]. A putative ZPS biosynthesis cluster in *P. goldsteinii* RV-01 was identified via searching the homologs of *wcfR* and *wcfS* (Figure 4). A match with the *wcfR* gene (gene 3065) was found in the genome of *P. goldsteinii* RV-01 (38.6% protein identity with those of *B. fragilis* NCTC9343), and a match with the *wcfS* gene (gene 3066) was found in the genome of *P. goldsteinii* RV-01 (69.3% protein identity with those of *B. fragilis* NCTC9343). This putative ZPS biosynthesis cluster also carried genes responsible for the glycosyltransferase, polymerase or ligase, and flippase of capsular polysaccharides (Figure 4). Thus, *P. goldsteinii* RV-01 may produce ZPS with anti-inflammatory properties.

#### 2.3.4. Bioinformatic Analyses for Immune Regulatory Proteins

A previous study has highlighted that cytoplasmic β-N-acetylhexosaminidase, after DC-T antigen presentation, enhances the cell population abundance of CD4+ CD8αα+ intraepithelial lymphocytes (CD4IELs) in the intestines [21]. In addition, an outer membrane protein, OmpA1—which increased the production of IL-22 and Reg3γ proteins that show anti-bacterial effects in the intestine—is involved in intestinal homeostasis [19]. Searching the whole genome of *P. goldsteinii* RV-01 revealed that a protein (gene 1081) was 97.3% identical with the hexosaminidase and harbored its immunostimulatory YKGSRVWLN epitope (Figure 5A). Moreover, a match with OmpA1 (gene 4451) was also found in the genome of the *P. goldsteinii* RV-01 (99.6% protein identity with those of *P. goldsteinii* ASF519, Figure 5B). Therefore, *P. goldsteinii* RV-01 was expected to have the ability to enhance the CD4IELs and to induce the production of IL-22 in an animal host.

### 2.4. Toxicological Studies

#### 2.4.1. HEK293 Cytotoxicity Assay

An in vitro cytotoxicity test on human embryonic kidney 293 (HEK293) cells was first performed to see whether the autoclaved *P. goldsteinii* RV-01 may show any cytotoxicity potential. Different numbers of *P. goldsteinii* RV-01 were used to treat HEK293 cells with a multiplicity of infection (MOI) of 1, 10, 50, 100, 500, 1000, and 2000. Then, the cytotoxicity was determined by measuring the level of extracellular LDH released from damaged cells. The result shown in Appendix A indicates that no cytotoxicity was observed even in the MOI up to 2000.

#### 2.4.2. Bacterial Reverse Mutation Test (Ames Test)

A Salmonella reverse mutation test (Ames test) was conducted with the *P. goldsteinii* RV-01 ingredient using five strains of *Salmonella typhimurium* (TA97a, TA98, TA100, TA102, and TA1535), in accordance with the OECD Guideline for Testing of Chemicals (Sect. 4, No. 471) (Table 4) [40]. In this study, the *P. goldsteinii* RV-01 ingredient at concentrations of 5, 2.5, 1.25, 0.625, and 0.3125 mg/plate was subjected to the mutagenic test. All experiments were performed in triplicate, and the mean revertant colony counts are provided in Appendix A. The overall results did not show any biologically relevant increases in the mean number of revertant colonies following exposure to the *P. goldsteinii* RV-01 ingredient compared with corresponding vehicle controls, in the presence or absence of metabolic activation (Appendix A). Therefore, *P. goldsteinii* RV-01 is non-mutagenic in the Salmonella reverse mutation assay.

#### 2.4.3. In Vitro Mammalian Chromosomal Aberration Test

This study was performed according to the OECD Guideline for Testing of Chemicals No. 473: In Vitro Mammalian Chromosomal Aberration Test (Table 4) [41]. The results revealed that no significant chromosomal aberrations were induced by the *P. goldsteinii* RV-01 ingredient compared with the negative control (Appendix A). In conclusion, the *P. goldsteinii* RV-01 showed a negative response to inducing chromosomal aberration.

#### 2.4.4. Rodent Micronucleus Test in Peripheral Blood

An in vivo micronucleus assay was conducted using human peripheral blood lymphocytes in accordance with the OECD Guideline for Testing of Chemicals (Sect. 4, No. 474) (Table 4) [42]. The main purpose of this study was to assess the possibility of the occurrence of micronuclei that may be caused by the *P. goldsteinii* RV-01 ingredient in the peripheral blood of rodents. The results for the number of micronucleated reticulocytes are provided in Appendix A. No significant difference in the number of reticulocytes with chromosomal structural aberrations (Appendix A) was observed between the treatment groups and negative control group. Therefore, the result of the micronucleus reaction of the *P. goldsteinii* RV-01 ingredient was considered to be negative. In brief, *P. goldsteinii* RV-01 did not induce structural or numerical chromosomal damage in human lymphocytes.

#### 2.4.5. Twenty-Eight-Day Repeated Dose Oral Subacute Toxicity Study in Rats 

To assess the potential adverse effects of the *P. goldsteinii* RV-01 ingredient, a 28-day dose range-finding oral toxicity assay in Sprague–Dawley (SD) rats (Table 4) was conducted in compliance with the Taiwan FDA GLP (2019), OECD-GLP (ENV/MC/CHEM (98) 17, 1997), and US FDA (21 CFR Part 58, 2023) on the principles of GLP for Nonclinical Laboratory Studies [43]. In this study, groups of five males and five females received 0 (control), 74 (4.0 × 10^9^ TFU/kg bw), 370 (2.0 × 10^10^ TFU/kg bw), or 1500 mg (8.109 × 10^10^ TFU/kg bw) of *P. goldsteinii* RV-01 once daily for 28 days. There were no deaths and no test item-related clinical signs. Animals administered the *P. goldsteinii* RV-01 ingredient showed similar body (Appendix A) and organ (Appendix A) weight changes. Further, no test items related to histopathological macroscopic or microscopic lesions were observed in the highest-dose group (Appendix A) nor any adverse effects in experimental animals. In conclusion, after 4 weeks of daily oral dosing of the *P. goldsteinii* RV-01 ingredient, the no-observed-adverse-effect level (NOAEL) was 1500 mg/kg/day in SD rats.

#### 2.4.6. Ninety-Day Repeated Dose Oral Subchronic Toxicity Study in Rats

This study was also conducted in accordance with OECD Principles of GLP Test Guidance (TG) No. 408 for a 90-day study (Table 4) [44]. The healthy SD rats (n = 10/sex/group; 6 weeks old) were administered *P. goldsteinii* RV-01 via gavage in doses of 0, 4.0 × 10^9^ (low dose), 2.0 × 10^10^ (medium dose), or 8.109 × 10^10^ (high dose) TFU/kg bw per day. No positive correlations in the degree and incidence rate of the histological lesions between the test article and control groups were noted (Appendix A). Furthermore, no clinical indications of test item-related toxicity were observed during this study. The functional observation battery did not identify any significant test item-dependent changes in behavior or in general ophthalmologic health. Based on the results of this study, the NOAEL was determined to be 8.109 × 10^10^ TFU/kg body weight/day, the highest dose tested in rats. The acceptable daily intake (ADI) is 900 mg/day based on an NOAEL of 1500 mg/kg/day, margin of safety (MOS) = 100, and 60 kg adult body weight. Taken together, the results of the repeat dose studies and the genotoxicity and mutagenicity studies show that the product is non-toxicogenic.

### 2.5. P. goldsteinii RV-01 Enhances Intestinal Immunity in Germ-Free (GF) Mice

GF mice were orally administered live *P. goldsteinii* RV-01 only. Then, the sera and colon tissues were collected after 2 weeks. Sera examination for liver and renal functions indicated that no adverse effects occurred (Figure 6A–C). Subsequently, the genes whose expression levels were affected by *P. goldsteinii* RV-01 alone in the intestinal tissue were characterized. *P. goldsteinii* RV-01 influenced the expression of multiple genes, with 255 upregulated genes and 557 downregulated genes identified in the colon tissue (Figure 6D). The pathways influenced by *P. goldsteinii* RV-01 were subsequently identified by GSEA (Figure 6E). Among these, the expression levels of genes related to the defense response to bacteria, antigen processing and presentation, T-cell differentiation, as well as B-cell homeostasis, were upregulated. By contrast, decreased expression levels of genes related to neuronal cell body and cell junction were also observed (Figure 6E). Previous results revealed that a normal immune development and a delayed neuronal response are elicited only after the microbiota accommodating homeostasis has been established in GF mice [45,46]. In this study, the administration of *P. goldsteinii* RV-01 significantly enhanced the immunity and significantly decreased the neuronal response in the colon, implying the involvement of *P. goldsteinii* in the development of immunity and neuronal signaling.

## 3. Discussion

The main purpose of this study was to assess the basic characteristics and evaluate the safety issue of the autoclaved bacterial strain *P. goldsteinii* RV-01 to be developed as a safe food supplement. Despite there being no *P. goldsteinii*-related adverse effects reported in previous studies, there were no data available nor any documented history of *P. goldsteinii*’s use as a food ingredient. Thus, given the vastly increasing amount of scientific research showing interest in proposing *P. goldsteinii* as a potential food ingredient, appropriate safety and toxicological evaluations were warranted. We used the safety assessment methodology following the tiered toxicity testing approach proposed by the EFSA and the US FDA guidance for assessing the safety of food ingredients. This approach involved the assessment of potential genotoxicity tests together with conducting an in vivo oral toxicity study in animals to assess the potential for subacute and subchronic toxicity. The results of the genotoxicity tests demonstrated that autoclaved *P. goldsteinii* is neither mutagenic (as assessed in the bacterial reverse mutation test) nor clastogenic or aneugenic (as evaluated in the mammalian cell micronucleus test and the mammalian chromosomal aberration test). Concordantly, in the 28-day and 90-day in vivo animal studies, there were no test item-related adverse effects on body and organ weights, food and water consumption, clinical observations and pathology, and neurobehavioral assessments. Furthermore, the NOAEL for the 90-day study was concluded to be 1500 mg/kg body weight/day (8.109 × 10^10^ cells/kg body weight/day), the highest dose tested. This NOAEL provides a safety factor of over 100-fold when compared with the worst-case anticipated exposure from use in foods and is the minimum safety requirement when applying animal experimentation data to humans [47]. The results therefore suggested that the autoclaved *P. goldsteinii* may have potential for application as a novel food supplement.

Currently, the majority of probiotics sold on the market mainly comprise microorganisms of the genera Lactobacillus and Bifidobacterium [48]. At the same time, continuing rapid advances in gut microbiota-related studies have led to significant progress in terms of exploring novel gut commensals’ functions as beneficial probiotics. An increasing number of new bacterial strains have been identified, isolated, and functionally characterized for the development of next-generation beneficial bacteria [49]. These NGPs might function as either novel food supplements, nutraceuticals, or as “live biotherapeutic products” [50,51]. Of these, *Anaerobutyricum soehngenii* (formerly *Eubacterium hallii*) [52], *Faecalibacterium prausnitzii* [53], *Akkermansia muciniphila* [54], *Christensenella minuta* [55], *Roseburia intestinalis* [56], *Bacteroides fragilis* [57], *Ruminococcus bromii* [50], and *P. goldsteinii* [3] (among others) were highlighted. These NGPs produce many postbiotic metabolites, leading to maintaining intestinal barrier integrity, modulating gastrointestinal and systemic immunity, increasing immunotherapy efficacy in cancer hosts, etc. [58]. Therefore, their presence and abundance in the gut might have impacts on maintaining health or even ameliorating various diseases [3,58].

*P. goldsteinii* is a gut commensal identified in the human intestines [2]. While the abundance of this bacterium is decreased by some chemicals or drugs such as aspirin [17], it was increased in animal hosts treated with prebiotic polysaccharides [6,7,19]. In recent years, an increasing amount of evidence has indicated that live *P. goldsteinii* is involved in maintaining the healthy gut lining and immunity, as well as decreasing low-grade smoldering inflammation in the intestines and increasing cellular mitochondrial and ribosomal activity [7,9]. Furthermore, the administration of *P. goldsteinii* was shown to be associated with the amelioration of several diseases [1,13,14,18]. There were some functional components identified in *P. goldsteinii*. In addition to SCFAs such as acetate and propionate (Figure 2A), *P. goldsteinii* also produced anti-inflammatory glycolipids (Pg-LPS) that antagonize TLR4 activity [9] and a cytoplasmic β-N-acetylhexosaminidase that enhances the cell abundance of CD4+ CD8αα+ intraepithelial lymphocytes (CD4IELs) in the intestines [21]. Furthermore, *P. goldsteinii* comprises an outer membrane protein, OmpA1, that increases the production of IL-22, leading to the production of anti-bacterial Reg3γ proteins in the intestine [19]. On the other hand, in collaboration with its hosts, *P. goldsteinii* produces a beneficial secondary bile acid, 7-α-LCA, that acts through antagonizing farnesoid X-receptor (FXR), leading to the facilitation of Wnt signaling and the self-renewal of intestinal stem cells [17]. In addition, *P. goldsteinii* and its extracellular vesicles (EVs) were reported to increase extracellular adenosine in the cecal lumen and induce IL-10-secreting dendritic cells in colonic lamina propria. Tolerogenic immunophenotypes ameliorating autoimmune disease are dependent on adenosine A2a signaling [59]. By contrast, no specific virulence- or allergy-related genes were identified, and only one *Tet*Q gene was highlighted in genomic screening (Appendix A). Interestingly, even though the *Tet*Q gene was identified in *P. goldsteinii* RV-01, *P. goldsteinii* RV-01 is sensitive to tetracycline administration (Table 3). Further study of the underlying mechanism is warranted.

Previous studies have indicated that various live NGP strains of the same species might work like double-edged swords, with some strains being advantageous, whereas others cause infections [3]. The potential risks of live NGPs comprise (i) translocation across the intestine and causing systemic infection, especially in vulnerable patients and pediatric populations; (ii) acquiring antibiotic resistance and virulence genes, followed by spreading thereafter; and (iii) interfering with the colonization of gut commensals in neonates [60]. To avoid these risks, there is increasing interest in the development of non-viable microorganisms (paraprobiotics) that are capable of functionally replacing the live bacteria. As the probiotic bacteria that have been killed are unable to replicate or colonize the human gut, they are generally regarded as safe. For example, the administration of up to 1 × 10^12^ inactivated *Bacteroides xylanisolvens* DSM 23964 did not cause significant clinical side effects [61]. On the other hand, the consumption of heat-killed *Mycobacterium setense manresensis* did not contribute to the pool of transmissible antimicrobial resistance genes [62]. Therefore, it was expected that the development of autoclaved *P. goldsteinii* would enforce the safety aspect. Although the bioinformatic analysis for potential virulence genes demonstrated that *P. goldsteinii* RV-01 is a non-pathogenic bacterium, safety concerns remained regarding the administration of viable bacteria to vulnerable individuals. Meanwhile, our results demonstrated that the cellular activity of autoclaved *P. goldsteinii* RV-01 was comparable with that of live bacteria (Appendix A), and autoclaved *P. goldsteinii* RV-01 retained an anti-inflammatory effect (Figure 3C). Therefore, the *P. goldsteinii* RV-01 was autoclaved before proceeding to the nonclinical safety assessment.

Dead probiotics may still maintain their beneficial effects, as the key active components may still be preserved in bacteria that have been killed [60,63]. Previous studies showed examples of heat-killed bacteria exhibiting similar therapeutic effects compared to those of live probiotics [64]. One example of this is that *Akkermansia muciniphila* retained its ability to provide trained immunity and to ameliorate airway allergies after being heat-killed at 70 °C for 15 min [65,66]. Other examples include the finding that killed *L. paracasei* maintained anti-aging functions [67], improved physical and cognitive activities, and extended the life span of *Caenorhabditis elegans* [68], as well as modulated brain activities [69]. In addition, the administration of heat-inactivated *Lactobacillus gasseri* tablets still lowered anxiety, improved sleep, and produced more n-valeric acid in young adults [70]. Further, the administration of a heat-killed *Lactobacillus helveticus* strain was still able to modulate anxiety-like or depression-like phenotypes in adolescent male mice [71]. Heat-killed *Enterococcus fecalis* maintained its activity in decreasing depressive and anxiety-like behaviors in mice [72]. Our recent study also indicated that the phosphorylation of CagA and the pathology this initiated were increased in *Helicobacter pylori*-infected cells, which was significantly decreased in cells pretreated with either live or heat-inactivated *P. goldsteinii* [13]. Our results demonstrated that the cellular activity of autoclaved *P. goldsteinii* RV-01 was comparable with that of live bacteria, and autoclaved *P. goldsteinii* RV-01 retained its anti-inflammatory effect. Thus, anti-inflammatory lipopolysaccharide and zwitterionic polysaccharides that may survive after autoclaving were speculated to constitute active components of autoclaved *P. goldsteinii* RV-01 ingredients. In brief, in addition to demonstrating a reduced risk of toxicity and increased stability, heat-killed probiotics may still be functional in promoting gut homeostasis and health [73].

In conclusion, we genetically and phenotypically characterized *P. goldsteinii* RV-01 and conducted safety-associated studies of autoclaved *P. goldsteinii* RV-01. The results of in vitro, in vivo, and bioinformatic analyses indicate that autoclaved *P. goldsteinii* RV-01 does not have aberrant effects. Therefore, it was suggested that the autoclaved *P. goldsteinii* RV-01 is safe for use as a novel food ingredient.

## 4. Materials and Methods

### 4.1. Ethics Statement

The Ethics Committee approved all procedures and techniques used in this research (Taipei Medical University Hospital, IRB number N202210087). All animal experiments were conducted following the approval of the Institutional Animal Care and Use Committee (approval numbers: MG113008, MG113009, and MG113010) and performed according to the guidelines in an OECD/Good Laboratory Practice (GLP) for Nonclinical Laboratory Studies-certified facility.

### 4.2. Isolation of P. goldsteinii RV-01

*P. goldsteinii* RV-01 was isolated from the feces of a healthy adult by streaking on anaerobic blood agar plates, Bacteroides Bile Esculin agar plates, and Laked blood with kanamycin and vancomycin agar plates (Creative, New Taipei city, Taiwan). Bacteria were grown at 37 °C in a Whitley A35 anaerobic chamber (Don Whitley, Bingley, UK) with mixed anaerobic gas (5% carbon dioxide, 5% hydrogen, 90% nitrogen), and the anaerobic condition was confirmed using an anaerobic indicator (Oxoid, Hampshire, UK). Colonies were picked and screened by PCR using *P. goldsteinii*-specific 16S rRNA primers [7] and then confirmed by sequencing the full-length 16S rRNA gene via Sanger’s method.

### 4.3. Production of P. goldsteinii RV-01 Ingredient

The production of *P. goldsteinii* RV-01 by fermentation was conducted according to Good Manufacturing Practice (GMP) and Hazard Analysis and Critical Control Point conditions. A 16S rRNA gene sequencing-validated working cell bank (WCB) maintained at the production facility was used to store inoculum for culture initiation. Briefly, a seed culture was initiated using an aliquot from the WCB, and the seed culture was then used to inoculate the production culture in the fermentation vessel. The growth of bacteria was achieved under 37 °C for 24 h. Cells were separated from the growth medium by pellet centrifugation and then subjected to autoclaving (121 °C for 15 min). After mixing the autoclaved cells with maltodextrin (1:6, *w*/*w*), followed by freeze-drying in a lyophilizer, a cake was obtained and then ground to powder. The enumeration of *P. goldsteinii* RV-01 in the powder was conducted via flow cytometry using a LIVE/DEAD^®^ BacLight™ Bacterial Viability and Counting Kit (Invitrogen, Carlsbad, CA, USA) [74,75]. This method, which is based on membrane integrity [76], can be used to monitor the number of dead and intact cells represented as total fluorescent units (TFUs). In the specific use described here, total cells of *P. goldsteinii* RV-01 are defined as TFUs, which represent the sum of dead cells. The TFUs/g of *P. goldsteinii* RV-01 ingredient for toxicologic studies was determined to be 5.406 × 10^10^.

### 4.4. Genome Sequencing and Analysis

The genomic DNA of *P. goldsteinii* RV-01 was extracted by using an Easy Pure Genomic DNA Spin Kit (Bioman, New Taipei City, Taiwan). The long fragment sequences generated by the third-generation Oxford Nanopore sequencing platform were initially assembled using Flye (version 2.9.1)/Canu (version 2.2) [77,78]. The preliminary Canu contigs then used Racon (version 1.5.0) for self-sequence correction [79]. Medaka (version 1.11.11.8.0) was used to correct the sequences of Flye contigs or Racon-corrected contigs again according to the machine learning model, and then homopolish (version 0.4.1) [80] was used to make the final sequence correction with the sequences of similar species in the database. After completing the sequence base correction of the contigs, Prokka (version 1.14.6) [81] was used to predict the position of the gene on the sequence. Subsequently, functional annotations were performed against gene annotation databases (COG, CARD, VFDB, dbCAN, and Allermatch). Finally, Circos (version 0.69.9) was used to draw a genome map based on the information obtained.

### 4.5. Determination of Minimum Inhibitory Concentration (MIC)

The MIC of the antimicrobials in *P. goldsteinii* was determined via the broth microdilution method and according to the internationally recognized standard of the Clinical and Laboratory Standard Institute (CLSI; www.clsi.org, accessed on 9 August 2024) for anaerobic bacteria. We tested the susceptibility of *P. goldsteinii* RV-01 to a selection of 14 antibiotics, including ampicillin, vancomycin, gentamycin, kanamycin, streptomycin, erythromycin, clindamycin, tetracycline, chloramphenicol, tylosine, apramycin, nalidixic acid, sulfonamide, and trimethoprim. The *P. goldsteinii* MTS01 strain, *P. goldsteinii* JCM13446 (ATCC BAA-1180) strain, *P. distasonis* ATCC8503 strain, and *P. merdae* ATCC43184 were also included as comparators. *B. fragilis* ATCC25285 was utilized as a positive control, as suggested by CLSI.

### 4.6. Quantification of Short-Chain Fatty Acid Production

Aliquots of 200 µL of culture supernatant from three independent fermentations or culture medium control were mixed with 70 µL of water, 50 µL of a 20 mM NaOH solution, and 160 µL of chloroform. The mixture was shaken vigorously for 1 min and then centrifuged at 1200× *g* for 15 min to obtain the upper layer. An amount of 200 µL of the sample was added to 70 µL of internal standard (C4-d8), 80 µL of isobutanol, and 100 µL of pyridine. The mixture was thoroughly shaken. Next, 50 µL of isobutyl chloroformate was added, followed by shaking for 2 min and subsequent sonication for 5 min. After esterification, 100 µL of hexane was added for extraction. The mixture was shaken and then centrifuged at 1200× *g* for 5 min. The upper layer was collected for analysis by using the Bruker GC-MS System consisting of a Bruker 436GC coupled to a Bruker EVOQ Mass Spectrometer (Brucker, Bremen, Germany). Then, the data were collected using Bruker MSWS 8.2 Software.

### 4.7. Cell Activity

HEK-blue-mTLR2 cells (InvivoGen, San Diego, CA, USA) were treated for 20 h with heat-treated (100 °C for 15 min), autoclaved (121 °C for 15 min), or pasteurized (70 °C for 30 min) *P. goldsteinii* RV-01 with a multiplicity of infection (MOI) of 10. The NF-κB activation activity was determined by measuring the OD630nm per the manufacturer’s instructions.

HCT116-Dual cells (InvivoGen, San Diego, CA, USA) were treated with either autoclaved *E. coli* ATCC25922 or *P. goldsteinii* RV-01 alone for 20 h. In parallel, cells were also pretreated with autoclaved *P. goldsteinii* RV-01 for 2 h, followed by 20 h of autoclaved *E. coli* ATCC25922 treatment. The NF-κB activation activity was determined by measuring the OD630nm per the manufacturer’s instructions.

### 4.8. Cytotoxicity Assay

Human embryonic kidney 293 (HEK293) cells were treated for 22 h with *P. goldsteinii* RV-01 ingredients under an MOI of 1, 10, 50, 100, 500, 1000, and 2000. The cytotoxicity was measured using a lactate dehydrogenase (LDH) assay kit (Promega, Madison, WI, USA); then, the cytotoxicity (%) was calculated per the manufacturer’s instructions.

### 4.9. Toxicologic Studies

The list of toxicological studies related to the safety of *P. goldsteinii* RV-01 is shown in Table 4. The tests, carried out on rats, mainly comprised three genotoxicity and mutagenicity studies, one in vivo subacute toxicity study, and one in vivo subchronic toxicity study. All tests were performed following the procedures provided in the standard protocols listed in Table 4 [40,41,42,43,44].

### 4.10. Colonization in Germ-Free Mice

Germ-free mice were orally administered live *P. goldsteinii* RV-01 (5 × 10^8^ CFU) once at 8 weeks old (n = 3 in GF group; n = 4 in GF + RV-01 group). After 2 weeks of mono-colonization, the mice were sacrificed, and their serum and colon tissues were collected for biochemical analysis and bulk RNA transcriptomic analysis, respectively.

### 4.11. Transcriptomic Analysis

Total RNA was prepared from the colon tissues of mice (n = 3 in each group) by using the GENEzol™ TriRNA Pure Kit (Geneaid, New Taipei City, Taiwan). RNA sequencing was conducted in an Illumina HiSeq4000 using a paired-end run (2 × 150 bases). The raw data were first checked for sequencing quality using FastQC (version 0.12.1). Sequences that passed quality management were mapped to the *Mus musculus* reference genome using HISAT2 (version 2.1.0), and featureCount (version 2.0.0) was used to count the number of sequences. DESeq2 (version 1.26.0) [82] was used for differential gene screening, and clusterProfiler (version 3.14.3) was used for enrichment analysis of GSEA for gene ontology (GO) gene sets. The results of the analysis were plotted using R software (version 4.3.1).

### 4.12. Statistical Analysis

Data are shown as the mean ± standard deviation (SD). For statistical analysis of studies with more than two groups, data were analyzed using the one-way ANOVA method followed by Dunnett’s test for comparison between groups. For statistical analysis of studies with two groups, data were analyzed using Student’s *t*-test for comparison between groups. A probability value of less than 0.05 (*p* < 0.05) was considered as significant.

## Figures and Tables

**Figure 1 ijms-25-12660-f001:**
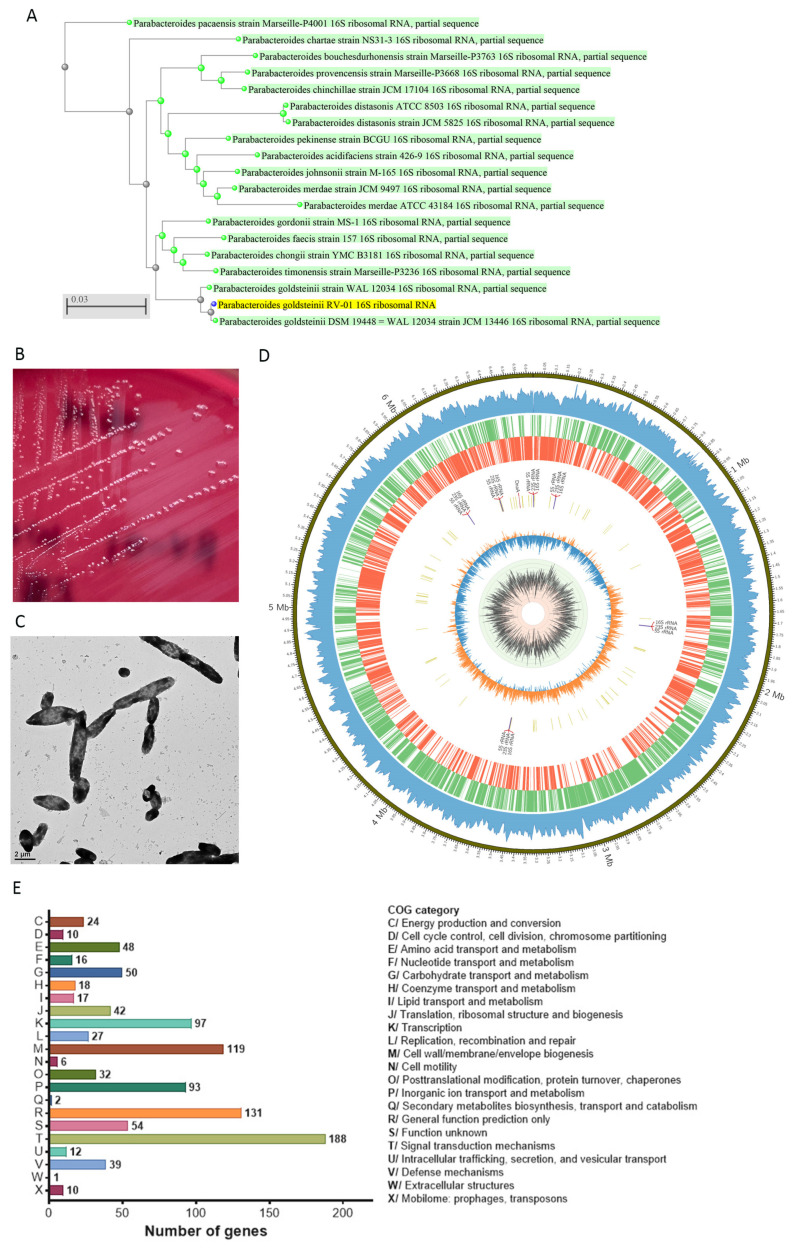
Characteristics of *P. goldsteinii* RV-01 strain. (**A**) The neighbor-joining phylogenetic tree between *P. goldsteinii* RV-01 and similar species and strains. (**B**) The colony morphology of *P. goldsteinii* RV-01 on an anaerobic blood agar plate. (**C**) The rod-shaped morphology of *P. goldsteinii* RV-01 under a transmission electron microscope. (**D**) The circular genome map of the *P. goldsteinii* RV-01 strain. The first (outermost) circle represents the locations of the chromosome. The second circle represents the read coverage. The third circle represents the predicted genes on the direct strand, and the fourth circle represents the predicted genes on the complementary strand. The fifth and sixth circles represent the labels and locations of 5S, 16S, 23S rRNA, and DnaA. The two innermost circles indicate the GC skew and GC content, respectively. (**E**) The functional annotation of the *P. goldsteinii* RV-01 genes analyzed against the COG database.

**Figure 2 ijms-25-12660-f002:**
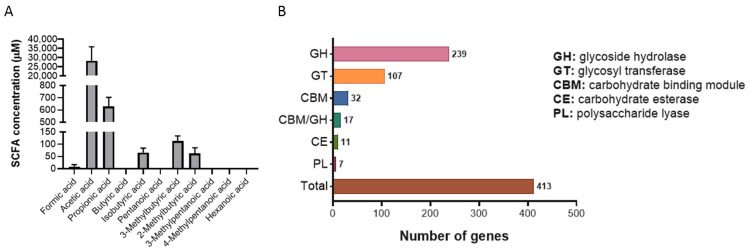
Production of SCFAs and bioinformatic analysis of CAZy in *P. goldsteinii* RV-01. (**A**) The levels of SCFAs in *P. goldsteinii* RV-01 culture supernatant analyzed by using MS spectrometry. (**B**) The carbohydrate utilization enzymes of *P. goldsteinii* RV-01 analyzed against Carbohydrate-Active Enzyme (CAZy) database.

**Figure 3 ijms-25-12660-f003:**
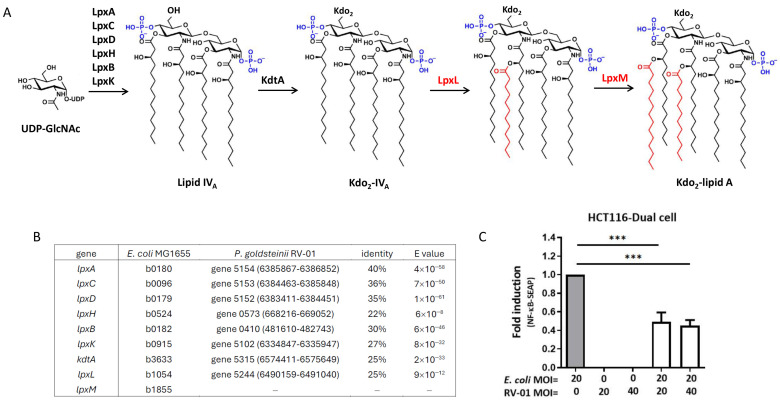
The predicted structure and NF-κB activation of *P. goldsteinii* RV-01 LPS. (**A**) The biosynthesis pathway of kdo2-lipidA in *E. coli*, indicating LpxL and LpxM are responsible for adding the fifth and sixth acyl chains to *E. coli* lipid A. (**B**) The putative genes for kdo2-lipid A synthesis of *P. goldsteinii* RV-01 were identified by BLAST searches using the *E. coli* MG1655 lipid A biosynthesis genes as queries. The gene ID and locations, as well as identities, in comparison with those in *E. coli* MG1655 are indicated. (**C**) The NF-κB activation of HCT116-Dual cells treated with *E. coli* or *P. goldsteinii* RV-01 alone, and treated with *E. coli* after pretreatment with *P. goldsteinii* RV-01. MOI, multiplicity of infection. ***, *p* < 0.001.

**Figure 4 ijms-25-12660-f004:**
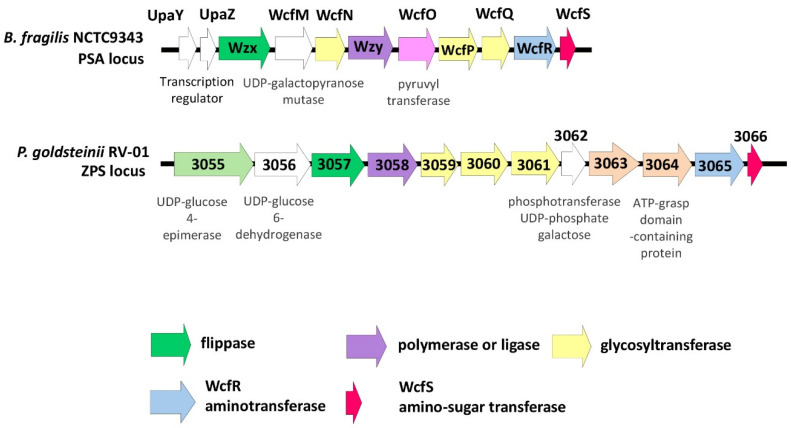
Genetic structure of a putative zwitterionic polysaccharide (ZPS) biosynthesis cluster in *P. goldsteinii* RV-01. A putative ZPS biosynthesis cluster in *P. goldsteinii* RV-01 was identified via searching the homologs of *wcfR* (indicated by a blue arrow) and *wcfS* (indicated by a red arrow), and the genetic structure was compared with those of *B. fragilis* NCTC9343. Genes important for the biosynthesis of capsular polysaccharides, such as glycosyltransferase (indicated by a yellow arrow), polymerase or ligase (indicated by a purple arrow), and flippase (indicated by a green arrow), are also shown.

**Figure 5 ijms-25-12660-f005:**
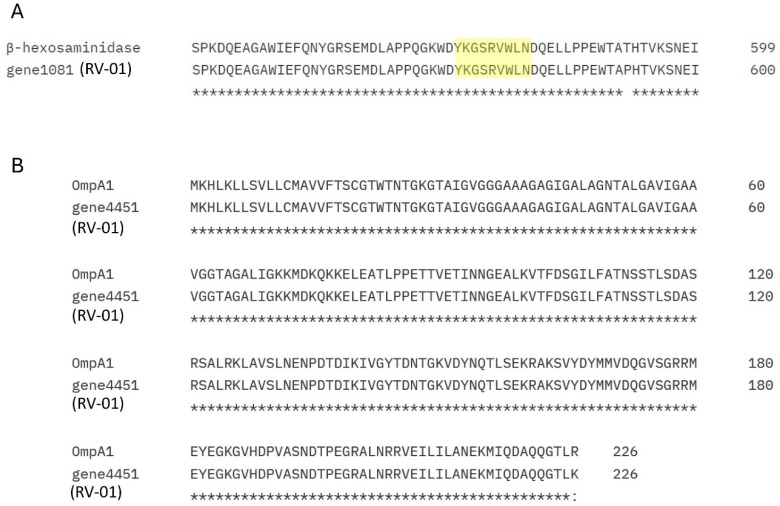
Bioinformatic analyses for functional proteins in *P. goldsteinii* RV-01. (**A**) Searching the whole genome of *P. goldsteinii* RV-01 revealed that a protein (gene 1081) was 97.3% identical with the hexosaminidase (WP_009860534.1) and harbored the hexosaminidase immunostimulatory YKGSRVWLN epitope (labeled by yellow color) [21]. (**B**) A match with the OmpA1 (gene 4451) was also found in the genome of *P. goldsteinii* RV-01 (99.6% protein identity with those of *P. goldsteinii* ASF519) [19].

**Figure 6 ijms-25-12660-f006:**
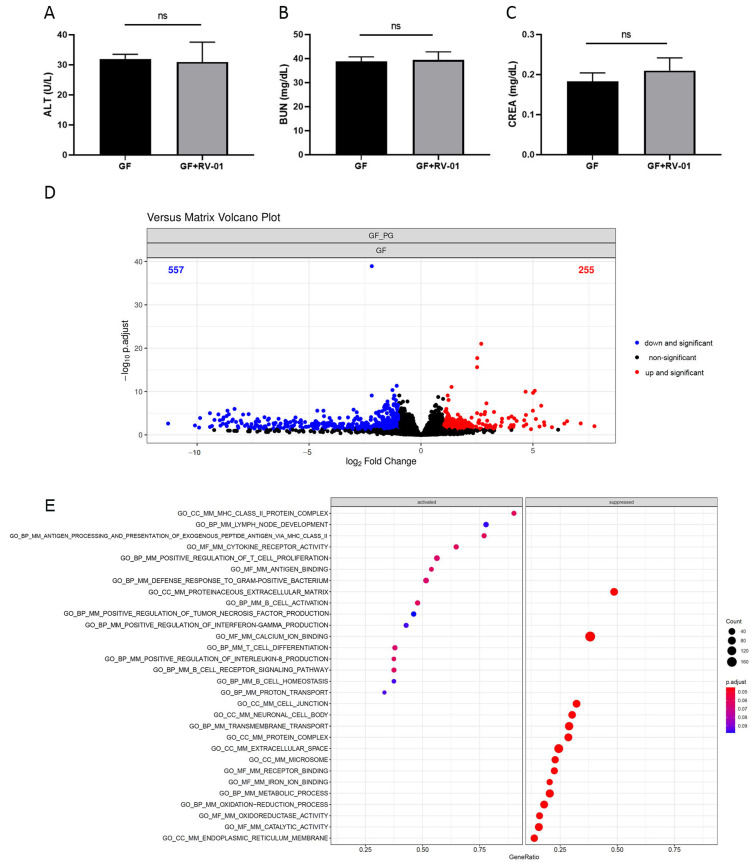
The effect of oral administration of *P. goldsteinii* RV-01 on GF mice. GF mice (n = 3 in GF group; n = 4 in GF + RV-01 group) were orally administered *P. goldsteinii* RV-01; then, the sera and colon tissues were collected after 2 weeks. (**A**–**C**) Serum levels of alanine aminotransferase (ALT), blood urea nitrogen (BUN), and creatinine (CREA). (**D**) The influence of *P. goldsteinii* RV-01 on gene expression in the colon of germ-free mice is indicated by volcano plot. (**E**) The pathways influenced by *P. goldsteinii* RV-01 determined using gene set enrichment analysis (GSEA) are shown in a dot plot. ns, not significant.

**Table 1 ijms-25-12660-t001:** Proximate composition of *P. goldsteinii* RV-01 ingredients.

Parameter	Typical Range	Method of Analysis
Protein	2.5–3.0 g/100 g	AOAC 992.15 (1992)
Fat	0.4–0.9 g/100 g	CNS5036
Ash	0.2–0.3 g/100 g	CNS5034
Carbohydrates	95.1–96.3 g/100 g	100 − (protein + fat + moisture + ash)
Energy value	397.8–401.7 Kcal/100 g	[4 × carbohydrates + 4 × protein + 9 × fat] Kcal

AOAC = Association of Official Analytical Chemists; CNS = Chinese National Standards.

**Table 2 ijms-25-12660-t002:** Average nucleotide identity (ANI) analysis between *P. goldsteinii* RV-01 and similar species and strains.

	ANI Value (%)					
	*P. goldsteinii* MTS01	*P. goldsteinii* DSM19448	*P. distasonis* ATCC8503	*P. merdae* ATCC43184	*P. gordonii* DSM23371	*P. faecis* DSM102983
*P. goldsteinii* RV-01	98.17	98.44	73.83	76.77	82.57	83.96

**Table 3 ijms-25-12660-t003:** Minimum inhibitory concentration (MIC) test results for 14 antibiotics.

Antibiotic Agent	MIC Range (μg/mL)						CLSI 2020 MIC Breakpoint for S (μg/mL)
	*B. fragilis* ATCC25285	*P. goldsteinii* RV-01	*P. goldsteinii* MTS01	*P. goldsteinii* JCM13446	*P. distasonis* ATCC8503	*P. merdae* ATCC43184	
Ampicillin	128	4	16	16	2	4	≤0.5
Vancomycin	32	16	>256	32	>256	32	NA
Gentamycin	>256	>256	>256	>256	>256	>256	NA
Kanamycin	>256	>256	>256	>256	>256	>256	NA
Streptomycin	>256	>256	>256	>256	>256	>256	NA
Erythromycin	8	8	8	8	1	8	NA
Clindamycin	0.5	2	4	0.25	<0.125	0.25	≤2
Tetracycline	0.25	4	0.5	0.25	0.25	8	≤4
Chloramphenicol	8	4	8	8	4	4	≤8
Tylosine	2	0.5	1	0.5	0.5	0.5	NA
Apramycin	>256	>256	>256	>256	>256	>256	NA
Nalidixic acid	256	128	128	128	128	128	NA
Sulfonamide	>256	>256	>256	>256	>256	>256	NA
Trimethoprim	32	>256	>256	>256	128	>256	NA

CLSI: Clinical & Laboratory Standards Institute. NA, not available.

**Table 4 ijms-25-12660-t004:** The list of toxicological studies with *P. goldsteinii* RV-01.

Test	Type of Study	Test System	Dose
1	Bacterial reverse mutation test (GLP, OECD test No. 471) [40]	*S*. Typhimurium TA97a, TA98, TA100, TA102, and TA1535	Up to 5 mg/plate (in absence and presence of S9 mix)
2	In vitro mammalian chromosomal aberration test (GLP, OECD test No. 473) [41]	Chinese hamster ovary cells (CHO-K1, BCRC No. 60006)	Up to 5 mg/mL for 3 h (in the absence and presence of S9 mix); up to 5 mg/mL for 18 h (in the absence of S9 mix)
3	Rodent micronucleus test in peripheral blood (GLP, OECD test No. 474) [42]	Imprinting control region mice (ICR mice)	Up to 2000 mg/kg
4	28-day repeated dose oral toxicity (GLP, OECD 407) [43]	Sprague–Dawley rats	Up to 1500 mg/kg bw per day (8.109 × 10^10^ TFU/kg bw per day)
5	90-day repeated dose oral toxicity (GLP, OECD TG408) [44]	Sprague–Dawley rats	Up to 1500 mg/kg bw per day (8.109 × 10^10^ TFU/kg bw per day)

## Data Availability

All relevant data are contained within the article. The original contributions presented in the study are included in the article/Appendix A. Further inquiries can be directed to the corresponding authors.

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
