# Peer review of "Characterization and Safety Evaluation of Autoclaved Gut Commensal Parabacteroides goldsteinii RV-01"

_ijms, 2024, doi:10.3390/ijms252312660_

Round 1
Reviewer 1 Report
Comments and Suggestions for Authors
This manuscript is on characterization and safety evaluation of autoclaved gut commensal Parabacteroides goldsteinii RV-01. The authors have confirmed in previous papers that Parabacteroides goldsteinii RV-01 plays an important role when chronic inflammation improves. Therefore, the authors confirmed the safety of RV-01 by conducting a 28-day subacute toxicity test and a 90-day subchronic toxicity test using rats in this study. This paper contains some valuable information, but there is also something to add for the benefit of the reader.
Point-1. Point - 1. The abstract should consist of objectives, materials and methods, key findings, and a conclusion or summary. So the authors should include them all. It should be rewritten around important information at a glance.
Point - 2. The authors fermented P. goldsteinii RV-01 and then sterilized and powdered the strains. However, the activity of parabiotics (inactivated RV-01) varies depending on the temperatures at which they are inactivated, and the authors need to describe in detail how they were sterilized. And did the authors inactivate RV-01 at different temperatures to confirm its activity?
Point - 3. The authors isolated P. goldsteinii RV-01 from feces using agar containing Kanamycin and vancomycinr. This means that RV-01 has an antimicrobial resistance gene. Of course, several antimicrobial resistance genes have also been identified in the analysis of CARD. Have you identified antibiotic-resistant genes in the autoclaved RV-01 powder?
Point - 4. Why didn't the authors perform an acute safety test to confirm the LD50 of autoclaved RV-01 powder or concentrated supernatant after RV-01 fermentation?
Point - 5. Did you confirm the production of gelatinase enzymes in RV-01 fermentation and the hemolysis in blood media?
Point-6. In Materials and Methods, authors have to describe the statistical analysis on the results. It is very important thing.
Author Response
Comments and Suggestions for Authors
This manuscript is on characterization and safety evaluation of autoclaved gut commensal Parabacteroides goldsteinii RV-01. The authors have confirmed in previous papers that Parabacteroides goldsteinii RV-01 plays an important role when chronic inflammation improves. Therefore, the authors confirmed the safety of RV-01 by conducting a 28-day subacute toxicity test and a 90-day subchronic toxicity test using rats in this study. This paper contains some valuable information, but there is also something to add for the benefit of the reader.
Point-1. Point - 1. The abstract should consist of objectives, materials and methods, key findings, and a conclusion or summary. So the authors should include them all. It should be rewritten around important information at a glance.
Response and revision: Lines 12-29: We have rewritten the abstract following the IJMS guidance of structured format which was without headings.
Point - 2. The authors fermented P. goldsteinii RV-01 and then sterilized and powdered the strains. However, the activity of parabiotics (inactivated RV-01) varies depending on the temperatures at which they are inactivated, and the authors need to describe in detail how they were sterilized. And did the authors inactivate RV-01 at different temperatures to confirm its activity?
Response and revision: Lines 117-122 and Supplementary Figure 1: The results about activities of P. goldsteinii RV-01 inactivated at different temperatures were added as suggested.
The RV-01 was inactivated by heat-treated (100 °C for 15 min), autoclaved (121 °C for 15 minutes) or pasteurized (70 °C for 30 minutes). Activities of P. goldsteinii RV-01 inactivated at different temperatures were compared (Supplementary Figure 1). The cellular TLR2 activation activity of autoclaved RV-01 was higher than that of heat-treated or pasteurized RV-01 and was comparable with that of live RV-01. Therefore, autoclaving was adopted for the production of P. goldsteinii RV-01 ingredient.
Point - 3. The authors isolated P. goldsteinii RV-01 from feces using agar containing Kanamycin and vancomycin. This means that RV-01 has an antimicrobial resistance gene. Of course, several antimicrobial resistance genes have also been identified in the analysis of CARD. Have you identified antibiotic-resistant genes in the autoclaved RV-01 powder?
Response and revision: LKV agar plate containing kanamycin and vancomycin is recommended for use in isolation of anaerobic gram-negative bacteria including P. goldsteinii. Vancomycin is employed to inhibit the growth of gram-positive bacteria and Kanamycin to inhibit gram-negative facultatively anaerobic bacilli. MICs results indicated P. goldsteinii RV-01, another two P. goldsteinii strains and two Parabacteroides species all presented high resistance levels to Kanamycin and Vancomycin, indicating resistance to these antimicrobials may be considered intrinsic.
Based on the analysis of CARD, only 1 gene which showed a significant match to the tetracycline resistance gene tetQ met the threshold qualifications of >60% coverage and >70% identity (Supplementary file 1). There was a possibility that the DNA fragment of the complete tetQ gene may still present in the P. goldsteinii RV-01 ingredient. To see whether the tetQ might be destroyed in this ingredient, PCR followed by agarose gel electrophoresis was performed to detect the presence of tetQ. The result showed that the tetQ gene was not detected in the P. goldsteinii RV-01 ingredient (Supplementary Figure 2), excluding the potential for horizontal spreading of the tetQ gene.
Point - 4. Why didn't the authors perform an acute safety test to confirm the LD50 of autoclaved RV-01 powder or concentrated supernatant after RV-01 fermentation?
Response and revision: The toxicological studies with P. goldsteinii RV-01 ingredients were performed according to FDA guidance on toxicity testing required for safety assessment of a new nonabsorbable food ingredient. Therefore, we did not perform an acute safety test to determine the LD50 of P. goldsteinii RV-01 ingredients. However, the results of 28-day repeated dose of oral subacute toxicity study in rats may suggest that the LD50 of autoclaved RV-01 powder was at least more than 8.109×1010 TFU/kg bw.
Point - 5. Did you confirm the production of gelatinase enzymes in RV-01 fermentation and the hemolysis in blood media?
Response and revision: The amino acid sequences of gelatinase enzyme of Enterococcus faecalis were used to search against the genome sequences of P. goldsteinii RV-01. There was no homolog of gelatinase detected in the P. goldsteinii RV-01. Figure 1B showed P. goldsteinii RV-01 colonies on CDC anaerobic blood agar plates revealed no obvious hemolysis.
Point-6. In Materials and Methods, authors have to describe the statistical analysis on the results. It is very important thing.
Response and revision: Lines 632-637: The statistical analysis was added in the Materials and Methods section as suggested.
Data were shown as mean±standard deviation (SD). For statistical analysis of studies with more than two groups, data were analyzed using one-way ANOVA method followed by Dunnett’s test for comparison between groups. For statistical analysis of studies with two groups, data were analyzed using Student t-test for comparison between groups. A probability value of less than 0.05 (p<0.05) was considered as significant.
Reviewer 2 Report
Comments and Suggestions for Authors
The article deals with the safety of administering a new strain of the bacterium Parabacteroides goldsteinii, which is considered to have probiotic properties. The authors focus on the possibility of consumption of this bacterium as an autoclaved preparation for use as a possible food additive.
I have the following questions that are not clarified in the text of the manuscript:
1. It is obvious that autoclaving will make preparations containing Parabacteroides goldsteinii safe to take. Why go to such lengths to prove that living cells-free bacterium extracts are safe? What dangerous components did you expect and test the strain to contain: toxins, endotoxins, capsules, allergens - please list the components you want to demonstrate that you have secured by autoclaving. Suppose you are using autoclaving only to kill the non-spore-forming bacteria. In that case, it seems your experiments have a foregone conclusion and do not constitute proof of hypothesis.
2. The authors are confident that the strain is safe. But is it useful? How was that proved?
3. Why do you think that the killed cells of Parabacteroides goldsteinii would show a probiotic effect? All the probiotic properties you have studied: formation of CFA, formation of zwitterionic polysaccharides, assistance in the maturation and development of the immune system, β-hexosaminidase production are typical for the living cells. Which of these probiotic properties apply to autoclaved culture too? Some of the molecules synthesized by P. goldsteinii may survive autoclaving, but the enzymes do not, please specify.
3. In my opinion, the greatest merit of the work is the sequencing of the complete genome of a new strain of P. goldsteinii and the bioinformatic safety analysis. I believe that authors should highlight these analyses as they best fit the journal's scope. The work should include bioinformatic analysis of genes and traits potentially beneficial to human health.
4. Long links to documents should not be included in the text; it is more appropriate to have them in references, for example, lines 97-101.
5. In my opinion, the authors should correct all the "advertising" elements in the text.
Author Response
Comments and Suggestions for Authors
The article deals with the safety of administering a new strain of the bacterium Parabacteroides goldsteinii, which is considered to have probiotic properties. The authors focus on the possibility of consumption of this bacterium as an autoclaved preparation for use as a possible food additive.
I have the following questions that are not clarified in the text of the manuscript:
- It is obvious that autoclaving will make preparations containing Parabacteroides goldsteinii safe to take. Why go to such lengths to prove that living cells-free bacterium extracts are safe? What dangerous components did you expect and test the strain to contain: toxins, endotoxins, capsules, allergens - please list the components you want to demonstrate that you have secured by autoclaving. Suppose you are using autoclaving only to kill the non-spore-forming bacteria. In that case, it seems your experiments have a foregone conclusion and do not constitute proof of hypothesis.
Response and revision: Lines 493-499: Although bioinformatic analysis for potential virulence genes demonstrated P. goldsteinii RV-01 is a non-pathogenic bacterium, safety for administration of viable bacteria in vulnerable individuals remained concerned. Meanwhile, our results demonstrated that the cellular activity of autoclaved P. goldsteinii RV-01 was comparable with live bacteria (Supplementary Figure 1), and autoclaved P. goldsteinii RV-01 retained the anti-inflammatory effect (Figure 3C). Therefore, the P. goldsteinii RV-01 was autoclaved before proceeding to the nonclinical safety assessment.
- The authors are confident that the strain is safe. But is it useful? How was that proved?
Response and revision: Lines 257-264: The result shown in Figure 3C suggested that P. goldsteinii RV-01 reduced the TLR4 related inflammation in HCT116 cells. As mentioned in Materials and Methods section, autoclaved P. goldsteinii was used in this experiment. The statements have been modified to clarify “autoclaved P. goldsteinii RV-01 retained the anti-inflammatory effect”.
- Why do you think that the killed cells of Parabacteroides goldsteinii would show a probiotic effect? All the probiotic properties you have studied: formation of CFA, formation of zwitterionic polysaccharides, assistance in the maturation and development of the immune system, β-hexosaminidase production are typical for the living cells. Which of these probiotic properties apply to autoclaved culture too? Some of the molecules synthesized by P. goldsteinii may survive autoclaving, but the enzymes do not, please specify.
Response and revision: Lines 515-520: Our results demonstrated that the cellular activity of autoclaved P. goldsteinii RV-01 was comparable with live bacteria (Supplementary Figure 1), and autoclaved P. goldsteinii RV-01 retained the anti-inflammatory effect (Figure 3C). Thus, anti-inflammatory lipopolysaccharide and zwitterionic polysaccharides which may survive after autoclaving were speculated as active components of autoclaved P. goldsteinii RV-01 ingredients.
- In my opinion, the greatest merit of the work is the sequencing of the complete genome of a new strain of P. goldsteinii and the bioinformatic safety analysis. I believe that authors should highlight these analyses as they best fit the journal's scope. The work should include bioinformatic analysis of genes and traits potentially beneficial to human health.
Response and revision: Lines 142 and 228: Sub-heading “Genomic and phenotypic characteristics for safety evaluation” and “Bioinformatic analysis of genes and traits potentially beneficial to human health” were added in the Result section as suggested. The “Genomic and phenotypic characteristics for safety evaluation” section includes minimum inhibitory concentration (MIC) evaluation of antibiotics and bioinformatic analyses for potential antibiotic resistance and virulence genes as well as allergic protein. The “Bioinformatic analysis of genes and traits potentially beneficial to human health” section includes production of short chain fatty acids (SCFAs) and bioinformatic analyses for anti-inflammatory LPS, zwitterionic capsular polysaccharide as well as immune regulatory proteins.
- Long links to documents should not be included in the text; it is more appropriate to have them in references, for example, lines 97-101.
Response and revision: The links to documents were moved to the references as suggested.
- In my opinion, the authors should correct all the "advertising" elements in the text.
Response and revision: The statements with possible advertising elements have been corrected or deleted as suggested.
Round 2
Reviewer 2 Report
Comments and Suggestions for Authors
The authors have completed all the recommendations and answered the questions in detail. Most importantly, they have carefully corrected the text in accordance with the comments made.
I recommend publishing in the current form.